# Neural Text-to-Speech Synthesis for Võro

**Liisa Rätsep**    **Mark Fishel**
Institute of Computer Science
University of Tartu
{`liisa.ratsep, mark.fisel`}`@ut.ee`

## Abstract

This paper presents the first high-quality neural text-to-speech (TTS) system for Võro, a minority language spoken in Southern Estonia. By leveraging existing Estonian TTS models and datasets, we analyze whether common low-resource NLP techniques, such as cross-lingual transfer learning from related languages or multi-task learning, can benefit our low-resource use case. Our results show that we can achieve high-quality Võro TTS without transfer learning and that using more diverse training data can even decrease synthesis quality. While these techniques may still be useful in some cases, our work highlights the need for caution when applied in specific low-resource scenarios, and it can provide valuable insights for future low-resource research and efforts in preserving minority languages.

## 1 Introduction

The advancements in neural text-to-speech (TTS) technology have greatly improved the quality of speech synthesis for many languages. However, despite the potential benefits of TTS for facilitating accessibility and language preservation, developing TTS systems for low-resource languages remains challenging due to the limited availability of training data for these languages.

Võro, a Finno-Ugric minority language spoken in Southern Estonia, serves as a great example of a low-resource language that could benefit from TTS technology. While linguistic resources for Võro are limited, the language is closely related to Estonian – a high-resource Finno-Ugric language with significantly more datasets, tools, and pretrained models.

The goal of this paper is to present the first high-quality neural TTS system for Võro and evaluate various low-resource NLP techniques for improving synthesis quality for the language. By leveraging existing Estonian TTS models and datasets, we investigate the impact of transfer learning from related languages and multi-speaker and multilingual approaches on the TTS quality of Võro.

The main contributions of this paper are:

1. We develop the first high-quality neural text-to-speech system for Võro and make it publicly available[1].

2. We show that having only 1.5 hours of Võro speech data per speaker is sufficient to develop TTS systems for low-resource languages without using cross-lingual transfer learning or additional monolingual data.

3. We highlight the potential negative effects of diversifying low-resource TTS datasets with data from closely related languages.

## 2 Background

As neural text-to-speech models require vast amounts of data, existing research has proposed several approaches to mitigate the issue of insufficient training data. For example, several works have shown that cross-lingual pretraining improves the quality of low-resource TTS systems (Chen et al., 2019; Xu et al., 2020).

In a survey on multilingual strategies for low-resource TTS, Do et al. (2021) evaluated the usefulness of using multilingual datasets for improving low-resource language performance. They observed that for sequence-to-sequence models, including additional data from other languages is almost always beneficial and often overweighs the negative effect of having a lower ratio of target data in the entire training dataset. The authors also noted that there is no clear evidence that

---

[1] `https://neurokone.ee`

using supporting languages from the same language family is more beneficial but claimed that using a shared input representation space (such as phonemes) may be more important.

At the same time, using closely related languages to boost low-resource performance has been successfully used for many text-based NLP tasks, including for developing Finno-Ugric machine translation systems that also include the Võro language (Tars et al., 2021). Unfortunately, the usage of neural methods for Võro has so far been limited to this example. There is also no existing research on Võru TTS. While the Estonian Language Institute and the Võro Institute have collaborated to create an HMM-based TTS system for Võro[2], this work has not been described in research.

## 3 Methodology

In this section, we present our methodology and experiment setup. Our approach evaluates the benefits of low-resource TTS approaches when training non-autoregressive Transformer-based models (Ren et al., 2019; Łańcucki, 2021). We focus on three common strategies – cross-lingual transfer learning from a pre-trained Estonian TTS model, combining data from multiple Võro speakers, and including Estonian data to create a multilingual system. Additionally, we explore data augmentation to handle the orthographic variation of Võro.

### 3.1 Datasets

Our experiments used speech data from two Võro speakers – an adult male and a child (female). Both datasets were attained from the Estonian Language Institute and contained an identical set of 1132 sentences, out of which 100 were set aside for evaluation purposes.

The Estonian dataset consisted of 6 male and 4 female speakers from the Speech Corpus of Estonian News Sentences (Fishel et al., 2020) and the Estonian Language Institute's audiobook corpora (Piits, 2022a,b). A subset of 1000 sentences per speaker was selected from the Estonian corpora to balance the training dataset.

The audio files were resampled at 22050 Hz and converted into mel-spectrograms using a Hann window with a frame size of 1024 and a hop length of 256. The mel-spectrogram frames were

aligned to the graphemes using the Estonian alignment model by Alumäe et al. (2018). Training a separate alignment model for Võro was also considered, but initial testing showed that the Estonian model was successfully able to produce high-quality alignments. The alignment was also used to trim excessive pauses in the audio.

All datasets were lowercased, and punctuation was normalized to a limited set of characters to reduce the vocabulary size. In total, the training dataset contained 3 hours of Võro and 14 hours of Estonian speech.

### 3.2 Data Augmentation

While the Võro dataset follows a standardized version of Võro orthography, many speakers and well-known news outlets do not conform to this standard. For example, the glottal stop ($q$) may be omitted or used only when it affects the meaning of the word, and some speakers may also use an apostrophe instead the letter $q$. Similarly, an apostrophe or an acute accent that marks palatalization is often used only when it affects the meaning.

In order to create a system that could successfully synthesize speech from all common written formats of Võro, we considered this to be an important challenge. As there are no existing NLP tools for Võro that would allow us to analyze these features automatically, we decided to use data augmentation to generate orthographic alternatives where glottal stops or palatalization features were removed for the system to cope with different orthographies.

Additionally, while our dataset contained the letter $y$, all cases of it were replaced with $\tilde{o}$ as they are no longer differentiated according to the orthographic standardization changes from 2005.

### 3.3 Model Configuration

All models were trained using an open-source implementation[3] of a non-autoregressive Transformer-based (Vaswani et al., 2017) model. The architecture is similar to FastPitch (Łańcucki, 2021) with explicit duration and pitch prediction components. An existing multispeaker model for Estonian (Rätsep et al., 2022) was used for our cross-lingual transfer learning experiments. In multispeaker systems, the speaker identity was marked with a prepended global style token (Wang et al., 2018).

---

[2]https://www.eki.ee/~indrek/voru/index.php

[3]https://github.com/TartuNLP/TransformerTTS

We trained models with three different data configurations – single-speaker Võro models for each speaker, multi-speaker Võro models with both speakers, and multi-speaker multilingual models with both Estonian and Võro data. For each data configuration, we also trained another model, which was initialized using the weights of the existing Estonian model. All models were trained for at least 400k steps and using identical hyperparameters.

## 4 Results

To assess the quality of the models, we conducted a mean opinion score (MOS) (Chu and Peng, 2001) evaluation[4] among volunteers from the Võro community. The evaluators were required to know the Võro language but did not have to be native speakers. Of the 41 volunteers, 6 considered themselves native speakers, and 9 had a self-reported Võru level of C1 or higher. Many participants with lower levels of Võru knowledge also mentioned that their passive language skills were higher as they mostly used Võro when communicating with older family members who were native speakers.

The evaluation used a subset of 50 random sentences per speaker (100 total per method) from the held-out dataset, and the samples were generated using pretrained HiFiGAN (Kong et al., 2020) models. The appropriate model for each speaker was selected by evaluating samples generated with multiple vocoder models. For the lower-pitched male speaker, we used a model trained on the VCTK dataset (Yamagishi et al., 2019), and for the child speaker, we used a model trained on the LJ Speech (Ito and Johnson, 2017) corpus and finetuned on Tacotron 2 (Shen et al., 2018) output. We also included ground truth samples from the held-out dataset and ground truth samples converter to mel-spectrograms and reconstructed by the same vocoder models.

The evaluation results can be seen in Table 1. Expectedly, ground truth samples in their original and reconstructed forms scored the highest among the participants. From the TTS models, the highest scores were given to single-speaker models. These were followed by the multi-speaker Võro models, but the performance drop from the single-speaker models should not be considered signif-

| Method | MOS |
|---|---|
| Ground truth | $4.03 \pm 0.12$ |
| Ground truth + vocoder | $3.83 \pm 0.13$ |
| Single-speaker | $3.55 \pm 0.15$ |
| Single-speaker (transfer) | $3.62 \pm 0.15$ |
| Multi-speaker | $3.43 \pm 0.15$ |
| Multi-speaker (transfer) | $3.50 \pm 0.13$ |
| Multilingual | $3.10 \pm 0.15$ |
| Multilingual (transfer) | $3.29 \pm 0.15$ |

Table 1: Mean opinion scores with 95% confidence intervals on the held-out dataset.

icant. The multilingual models showed consistently worse performance compared to the monolingual models. Additionally, we observe minor benefits from using cross-lingual transfer learning.

In addition to scoring samples, participants were encouraged to comment on their overall impressions of speech quality and the evaluation process. Many expressed a positive surprise about synthesis quality and mentioned the presence of TTS artifacts, such as crackling, as their main evaluation criteria. Some participants also noted that while almost all samples were intelligible, they did not always sound like a native Võro speaker, especially when producing the glottal stop sound. Unfortunately, as the participants did not know which models produced which samples, further analysis would be needed to assess whether all models are equally prone to this issue and whether it can also be observed in ground truth examples.

## 5 Discussion and Future Work

Unexpectedly, our MOS evaluation results are in conflict with existing low-resource TTS literature that reports benefits from diversifying training data with samples from other speakers or related languages and from using cross-lingual transfer learning. This brings into question both the usefulness of these techniques as well as our approach.

Firstly, it could be argued that the observations about the low negative performance impact of data imbalance by Do et al. (2021) may not apply to non-autoregressive Transformer-based systems, as the study focused on other methods, such as recurrent or convolutional neural networks. Therefore, the performance drop in multilingual models could still be caused by an imbalance between the

---

[4]https://tartunlp.github.io/
TransformerTTS/nodalida2023/

two languages in the dataset. Alternatively, as our model size was dictated by the existing pretrained Estonian models, it may lack sufficient capacity to work in a multilingual setting.

Additionally, it is possible that we should no longer consider Võro a low-resource language in this task. Based on initial testing with Estonian datasets, we found that the required amount of speech data for Transformer-based models to produce coherent speech is between 1-2 hours, and improvements from using more data are significantly less noticeable. Similar observations about reduced data requirements for Transformer-based models have also been recently reported by Pine et al. (2022). In our case, we had 1.5 hours of speech per speaker, and it may have been sufficient for us not to benefit from additional data from other speakers. Alternatively, as the two Võro datasets contained identical sentences, they may not differ sufficiently to benefit from each other. However, a more detailed evaluation methodology could be considered to measure the effects on specific features of synthetic speech, such as prosodic variability or pronunciation mistakes.

As our work focused on creating a high-quality system for Võro without applying artificial constraints, such as using smaller subsets of the high-resource datasets, these points were not explicitly explored in our work. However, in the future, low-resource TTS strategies should be further reviewed specifically for Transformer-based architectures and for different levels of resource constraint. Until then, these strategies should be used with caution and evaluated for each specific low-resource scenario.

## 6 Conclusion

This article presented the first high-quality neural text-to-speech system for the Võro language. We explored the usage of Estonian TTS models and datasets to boost the performance of our low-resource use case.

Our results suggest that we can achieve high-quality Võro TTS without transfer learning or using data from multiple speakers or closely related languages. While these techniques may still be helpful in some cases, we highlight the need for further research and evaluation when applied in specific low-resource scenarios.

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
