# OpenReview forum: "Neural Text-to-Speech Synthesis for Võro"
_NoDaLiDa/2023/Conference — NoDaLiDa 2023_

### Official Review · Reviewer_EUBB · 2023-03-10
**Review of the paper "Neural Text-to-Speech Synthesis for Võro"**

**Rating:** 8
**Confidence:** 4

**Review:**

The paper introduces the neural TTS system for the Võro language. The work presented in this paper is a good example of how recent technological advances enable the development of speech technology applications for low-resourced  minority languages.
The use of state-of-art approaches with a rather small training corpus has resulted in quite high MOS values. For comparison, it would be good to present the amount of training data and MOS scores for standard Estonian TTS.
The claim that Võro should no longer be considered a low-resource language sounds too optimistic.
The paper is generally well-written and easy to follow.

Remarks:
- Lines 124-125: check the grammar of the sentence
- Line 328: ... for Transformer-based to produce... - Transformer-based model?
- Ch6, the first sentence remains unclear: "As our work focused on creating a high-quality system for Võro without applying artificial constraints, these points were not explicitly explored in our work". - what artificial constraints? which points are not explored?

**Paper Type:**

Short paper

---

### Official Review · Reviewer_dSDN · 2023-03-11
**TTS for Voro**

**Rating:** 7
**Confidence:** 5

**Review:**

The paper presents development and evaluation of TTS systems for the Voro language.  This is the first high-quality neural TTS system developed for the language and it presents interesting results where simply using 1.5 hours of Voro recordings is enough to get better quality then when using transfer learning from multiple speakers or closely related languages (Estonian).  The paper is clearly written and presents the work well (I noticed a couple of typos so make sure to proof-read the manuscript).  The material of the paper suits NoDaLiDa well and should be accepted in the conference.

**Paper Type:**

Short paper

---

### Decision · Program_Chairs · 2023-03-17

Accept